# Chromatographic Analyses of Spirulina *(Arthrospira platensis*) and Mechanism of Its Protective Effects against Experimental Obesity and Hepatic Steatosis in Rats

**DOI:** 10.3390/medicina59101823

**Published:** 2023-10-13

**Authors:** Fatma Arrari, Mohamed-Amine Jabri, Ala Ayari, Nouha Dakhli, Chayma Ben Fayala, Samir Boubaker, Hichem Sebai

**Affiliations:** 1Laboratory of Functional Physiology and Valorization of Bio-Resources, Higher Institute of Biotechnology of Beja, University of Jendouba, Beja 9000, Tunisia; amine.jabri@isbb.u-jendouba.tn (M.-A.J.); hichem.sebai@u-jendouba.tn (H.S.); 2Laboratory of Human and Experimental Pathological Anatomy, Pasteur Institute of Tunisia, Tunis 1002, Tunisia

**Keywords:** obesity, cafeteria diet, spirulina, hyperlipidemia, lipotoxicity

## Abstract

*Background and Objectives*: Obesity is currently a major health problem due to fatty acid accumulation and excess intake of energy, which leads to an increase in oxidative stress, particularly in the liver. The main goal of this study is to evaluate the protective effects of spirulina (SP) against cafeteria diet (CD)-induced obesity, oxidative stress, and lipotoxicity in rats. *Materials and Methods:* The rats were divided into four groups and received daily treatments for eight weeks as follows: control group fed a standard diet (SD 360 g/d); cafeteria diet group (CD 360 g/d); spirulina group (SP 500 mg/kg); and CD + SP group (500 mg/kg, *b.w*., *p.o.*) according to body weight (*b.w*.) per oral (*p.o.*). *Results:* Our results show that treatment with a CD increased the weights of the body, liver, and abdominal fat. Additionally, severe hepatic alteration, disturbances in the metabolic parameters of serum, and lipotoxicity associated with oxidative stress in response to the CD-induced obesity were observed. However, SP treatment significantly reduced the liver alteration of CD feed and lipid profile disorder associated with obesity. *Conclusions:* Our findings suggest that spirulina has a marked potential therapeutic effect against obesity and mitigates disturbances in liver function parameters, histological alterations, and oxidative stress status.

## 1. Introduction

Obesity is defined as an abnormal excess of body fat caused by a calorie consumption that exceeds energy expenditure [1]. Obesity leads to adipose tissue accumulation, which has a harmful impact on health. Hyperlipidemia is a growing public health challenge worldwide, with significant health and economic impacts [2]. Currently, ca. 2.5 billion individuals are overweight or obese around the globe.

Obesity is a well-known risk factor for debilitating chronic diseases such as cardiovascular disease, cancer, cerebrovascular disease [3,4], and hepatic steatosis [5]. Nonalcoholic fatty liver disease (NAFLD) has developed as the most common liver pathology worldwide, and is strongly linked to the presence of oxidative stress [6,7]. Disturbances in lipid metabolism induce hepatic lipid accumulation, which disturbs different reactive oxygen species (ROS) generators, such as the mitochondria, endoplasmic reticulum, and NADPH oxidase [8]. Our study was conducted on rats with nutritional obesity caused by a cafeteria diet. This obesogenic diet induced weight gain and impaired carbohydrate homeostasis associated with insulin resistance [9,10,11]. In addition, it is widely known that obesity is linked to an oxidative stress status characterized by the overproduction of free radicals [12]. An imbalance between pro-oxidants and antioxidants may be caused by an excess of reactive oxygen species (ROS) or antioxidant depletion [13]. These ROS may originate from nicotinamide adenine dinucleotide phosphate (NADPH) oxidase and the mitochondrial respiratory chain [14]. Obesity also provides a rapid source of adipokines and pro-inflammatory cytokines [15] and promotes a chronic hepatic inflammatory state [16], as well as metabolic syndrome.

The spread of overweight and obesity in the world has encouraged us to develop natural remedies to prevent this disease. These remedies present the same pharmacological efficacy as chemical-based drugs, with fewer side effects.

Natural products play an essential role in the development ofnew drugs for the treatment of obesity, such as chamomile (*Matricaria recutita* L.) [17].

Spirulina is a filamentous cyanobacterium and spiral microalgae [18,19]. It is a blue-green algae that contains many components, such as minerals, protein, fiber, phytochemicals, vitamins, fatty acids, and carbohydrates. These phenolic compounds have been demonstrated to have the potential to combat the detrimental effects of free radicals, which are harmful to our food systems and bodies [20]. Spirulina can be used as apowerful treatment against several diseases due to its significant therapeutic properties, such as antioxidant, anti-inflammatory, anticancer, antiviral, and antibacterial activities [21]. Several studies suggest that SP protects against obesity by regulating appetite, food absorption, gut microbiota, insulin resistance, oxidative stress [22,23,24], and inflammation [25,26].

The aim of this study was to evaluate the protective effect of SP against CD-induced obesity in a rat model, as well as against the associated hepatic steatosis and inflammation.

## 2. Materials and Methods

### 2.1. Chemicals

Diethyl ether, eosin (E), hematoxylin (H), formaldehyde, epinephrine, bovine catalase, butylated hydroxytoluene (BHT), 2-Thio-barbituric acid (TBA), pyrogallol, NaCl, methanol, chlorhydric acid (Hcl), ethylene tetra-acetic (EDTA), and hydrogen peroxide were purchased from sigma-aldrish Co., Munich, Germany.

### 2.2. Prepartion of Spirulina (SP)

The flakes of spirulina were purchased from the Tunisian company Eden Life (Kettana, Tunisia). They wereground into a powder using an electric blender before being placed into bottles and kept dry in the dark. We mixed 5 g of SP powder with 50 mL of distilled water by light shaking 1 h before the treatment.

### 2.3. Characterization of Phenolic Compounds of Spirulina by HPLC-DAD-ESI MS/MS Analysis

The spirulina extract was dissolved in methanol and analyzed by LC-MS/MS using an Agilent Series 1100 LC system (Agilent Technologies, Palo Alto, CA, USA). This system was equipped with a photodiode array detector (PDA) and a triple quadrupole mass spectrometer, Micromass Autospec UltimaPt (Kelso, UK), interfaced with an ESI ion source. Separation was achieved using a Superspher^®^ 100 (12.5 cm × 2 mm i.d., 4 mm, Agilent Technologies, Rising Sun, Maryland, MD, USA) at 45 °C [27].

The samples (20 µL) were eluted through the column with a gradient mobile phase consisting of A (0.1% acetic acid) and B (acetonitrile) at a flow rate of 0.25 mL min^−1^. The following multistep linear solvent gradient was employed: 0–5 min, 2% B; 5–75 min, 88% B; 75.1–90 min, 2% B.

PDA was detected in the 200–400 nm wavelength range, and the mass spectra were recorded in negative ion mode under the following operating conditions: capillary voltage, 3.2 kV; cone voltage, 115 V; probe temperature, 350 °C; ion source temperature, 110 °C. The spectra were acquired in the *m/z* range of 150–750 amu.

Identification of phenolic compounds was based on chromatography with known standards, when available. PDA and mass spectra were used to confirm the identity of compounds previously reported in the literature [28,29]. The content percentage of each component in the crude extract was determined by a UV chromatogram at 210 nm.

### 2.4. Preparation of Cafeteria Diet

A cafeteria diet (CD) increases the risk of obesity in rats. The components of the CD included a mixture, with 50% being cookies, cheese, salami, peanuts, chocolate, and chips in a proportion of 2:2:2:1:1:1, and with other 50% being a standard diet (mix/standard diet, *w*/*w*) given to each groups for two months [11,30]. The composition of the cafeteria diet (420 kJ/100 g) consisted of 23% energy from protein, 35% energy from carbohydrates, 42% energy from lipids and 9% moisture content [30]. The cafeteria diet induced hyperphagia followed by obesity due toits high-fat and high-calorie contents.

### 2.5. Animals and Treatment

Male Wistar rats (n = 32 body weight (BW) 180–200 g) were housed four per cage for a 7-day acclimation period. The animals were procured from the Society of Pharmaceutical Industries of Tunisia (SIPHAT, Ben Arous, Tunisia) and used following the guidelines of the local ethics committee of Tunisia University for the use and care of animals, as recommended by the National Institutes of Health (NIH). The animal experiments and treatments were approved by the Ethical Committee of Biomedical (CEBM) for the care and use of animals in Tunisia (ref: JORT472004/2020). The number of rats used in the study was minimized in accordance with the 3Rs guidelines for the humane treatment of animals [31], and following the International Council for Laboratory Animal Science (ICLAS) guidelines.

Animals were fed a standard pellet diet (Badr Utique-TN) and provided with free access to water ad libitum. They were housed in facilities with a controlled temperature (22 ± 2 °C) and humidity (60% of relative humidity) and a 12 h light–dark cycle. After the adaptation period, the rats were divided into four groups, each containing eight rodents, with two cages for each treatment group. Over a period of two months, animals were fed a standard diet (SD) (groups I and IV) or a cafeteria diet (360 g/day) (CD) (groups II and III) at 09:00 h. On the other hand, groups I and II received distilled water while groups III and IV were treated with SP (500 mg/kg. *b.w.*, *p.o*.) for eight weeks. The treatments were administered orally every 24 h. The body weight of the animals was measured every three days.

### 2.6. Biochemical Assessment of Liver Tissue

On the last day of our treatment, animals were sacrificed by decapitation after overnight fasting. The liver was rapidly excised, homogenized in phosphate-buffered saline, and centrifuged for 15 min at 10,000× *g* at 4 °C [32]. The resulting supernatant was stored at −80 °C for the determination of biochemical parameters, including protein content, H_2_O_2_, sulfhydryl group (-SH groups), reduced glutathione (GSH), and malondialdehyde (MDA), as well as antioxidant enzyme activities and lipid composition. Blood was collected and centrifuged at 3000× *g* for 15 min, and the plasma was stored at −20 °C for further biochemical analysis. The remaining liver tissue was preserved in 10% formalin for histological examination.

### 2.7. Metabolic Parameters

The lipid markers, including total cholesterol (TC) (cat. no. 21014, Biomaghreb, Ariana, Tunisia), high-density lipoproteincholesterol (HDL) (cat. no. 23025, Biomaghreb, Tunisia), low-densitylipoproteincholesterol (LDL) (cat. no. 24022, Biomaghreb, Tunisia), and triglyceride (TG) (cat. no. 29010, Biomaghreb, Ariana, A, Tunisia),were analyzed in both plasma and liver samples. Concentrations were measured using a SELECTRAPRO XL automatic biochemical analyzer with the commercially available kits obtained from Biomaghreb, Tunisia (ISO 9001 certificate).

### 2.8. Biochemical Analysis

Glycemia, direct bilirubin, aspartate amino-transferase (ASAT) and alanine aminotransferase (ALAT) were measured in plasma samples using commercially available diagnostic kits (Biomaghreb, Ariana, Tunisia).

### 2.9. Oxidative Stress Assessment

The protein contents were determined using the Hartree method with a slight modification of the Lowry method [33]. The level of MDA was evaluated following the Drapper and Hadley [34] protocol, which involves reacting MDA with thiobarbituric acid. Ellman’s method was employed to estimate the concentration of -SH groups [35], and the Sedlak and Lindsay method was used to measure the GSH level [36].

### 2.10. Antioxidant Enzyme Activity Assays

Antioxidant enzyme activity assays were conducted as follows: Catalase (CAT) activity was determined using the method proposed by Aebi [37]. Superoxide dismutase activity (SOD) was assessed using the epinephrin/adenochrome method according to Misra and Fridovich’s approach [38]. Glutathione peroxidase (GPx) activity was evaluated using the method described by Flohé and Günzler [39].

### 2.11. Determination of Reactive Oxygen Species

Dingeon’s method was employed to assess the level of hydrogen peroxide (H_2_O_2_) in the samples [40]. In the presence of peroxidase, hydrogen peroxide reacts with p-hydroxybenzoic acid and 4-aminoantipyrine, forming a quinoneimine compound. The optical density was measured at 505 nm.

The level of hydroxyl radical was determined using the method of Paya et al. [41]. This involved the oxidation of desoxyribose by hydroxyl radicals generated by the Fe^3+^ascorbate-EDTA-H_2_O_2_pathway, followed by incubation with liver homogenate at 37 °C for one hour. The reaction was halted by adding TCA (2.8%) and TBA (1%), and boiled at 100 °C for 20 min. Changes in absorbance at 532 nm were measured compared to a blank containing desoxyribose and buffer.

The level of superoxide anions was determined using the Marklund method with minor modifications [42]. Briefly, the samples were incubated in tris-Hcl buffer, and pyrogallol was added to the reaction mixture, which was then incubated at 25 °C for 5 min. The reaction was stopped by adding Hcl and the absorbance at 420 nm was measured against the blank.

### 2.12. Histopathological Study

Immediately after sacrifice, the dissected rat livers were fixed in 10% paraformaldehyde and then embedded in paraffin [43]. Subsequently, the liver tissues were sectioned into 5 µm slices, deparaffinized, and rehydrated in ascending concentrations of ethyl alcohol (70–100%). Finally, the samples were stained with hematoxylin and eosin (H&E).

### 2.13. Assessment of Liver Cytokines

The levels of cytokines in the supernatant were determined using standard sandwich enzyme-linked immunosorbent assay (ELISA) kits (Cat. No. CRS-B002 and CEA-C010, Bioscience, San Diego, CA, USA) according to the manufacturer’s instructions and expressed in pg/mg protein.

### 2.14. Data Analysis

Statistical analysis was conducted using Statistica 13.0 data analysis software (TIBCO Software Inc., Palo Alto, CA, USA).Prior to analysis, all results were assessed for normality and homoscedasticity. The impact of different diets on weight gain, total abdominal fat, body weight, food intake, and biochemical parameters was assessed using a one-way ANOVA followed by a post hoc least significant difference (LSD) test. Differences between treatments were determined by a Student’s *t*-test and considered significant at *p* < 0.05. For the assessment of antioxidant markers and lipid peroxidation in each treatments group, based on the applied dietary experiments, the non-parametric Kruskal–Wallis test with Mann–Whitney post-hoc analysis was employed.

## 3. Results

### 3.1. HPLC-DAD-ESI-MS/MS Analysis of Spirulina

The HPLC-DAD-ESI-MS/MS analysis of spirulina enabled the identification of flavonoids such as catechin and quercetin, as well as phenolic acids like chlorogenic acid, syringic acid, and sinapic acid. Additionally, resorcinol or resorcin, or benzene-1,3-diol, the meta isomer of benzenediol, was identified as a diphenol (Table 1). The chromatographic elution profile of all identified phenolic compounds is depictedin Figure 1.

### 3.2. Effect of SP and CD on Body, Liver, Abdominal Fat Weight, Weight Gain, and Food Intake

Obesity was assessed principally through body weight, abdominal fat weight, and weight gain to evaluate the effect of SP on obesity induced by CD. Animals fed with a cafeteria diet (CD) exhibited significant increases in body, liver, and abdominal fat and food intake, as well as gaining weight, compared to those fed the standard diet (SD) (Table 2). Treatment with SP led to a significant decrease in all weight parameters compared to those fed with CD.

### 3.3. Effect of SP and a CD on Liver and Plasma Lipid Profiles

To investigate the effects of obesity on the hypolipidemic effect of SP, the levels of TC, TG, HDL, and LDL were assessed in both plasma and hepatic tissue. The supplementation of a CD significantly increased TC, TG, and LDL levels while lowering the plasma HDL levels (Table 3). Additionally, a CD led to elevated hepatic TC and TG levels, but did not significantly affect LDL or HDL compared to the standard diet group (SD). Moreover, SP treatment significantly restored all these metabolic parameters to their baseline levels in the CD group.

### 3.4. Effect of SP and a CD on Liver Function

The effect of SP and CD treatment on liver function was assessed and is shown in Table 4. We observed that hepatic injuries are associated with a significant increase in glycemia, ASAT, ALAT, and direct bilirubin in the CD group. SP treatment significantly alleviated all these metabolic alterations compared to the CD group.

### 3.5. Effect of SP and a CD on Liver Oxidative Stress

The cafeteria diet (CD) induced a significant alteration in the liver’s redox balance, as evidenced by the significant increase in lipid peroxidation (Figure 2A) in obese rats compared to those fed a standard diet. A significant depletion in non-enzymatic antioxidants such as -SH groups (Figure 2B) and reduced glutathione levels (Figure 2C) were observed. Conversely, SP treatment effectively mitigated these perturbations in the liver tissue when compared to the CD group (*p* < 0.05).

### 3.6. Effect of SP and a CD on Liver Antioxidant Enzyme Activity

As shown in Figure 3, a significant decrease in antioxidant enzyme activities, such as SOD (Figure 3A), CAT (Figure 3B), and GPx (Figure 3C), was found in the liver of obese rats compared to the control fed a standard diet. However, eight weeks of SP treatment significantly alleviated this inhibition compared to the CD group.

### 3.7. Effect of SP and a CD on Liver Reactive Oxygen Species (ROS)

Figure 4 shows how different treatments affected the synthesis of hepatic hydrogen peroxide (Figure 4A), hydroxyl radicals (Figure 4B), and superoxide anions (Figure 4C). The CD induced an overload of hepatic reactive oxygen species (ROS), whereas SP treatment significantly reduced the liver ROS production to the basic levels.

### 3.8. HistopathologicalStudy

A histological analysis with H&E staining was used to evaluate the impact of SP and a CD on the accumulation of fat droplets. The standard diet group exhibited a normal histological architecture (Figure 5A). SP administered rats had normal livers and no obvious changes were detected (Figure 5D). However, after 8 weeks of CD feeding, acute necrosis and steatosis were observed, with fat vacuoles and cell ballooning apparent in the liver compared to the standard diet group (Figure 5B). SP treatment reduced the vacuoles of fat droplets in the liver cells of CD-fed rats, resulting in a normal liver (Figure 5C).

### 3.9. Effect of SP and a CD on Liver Cytokine Levels

The effects of SP and CD on liver tumor necrosis factor (TNFα) and interleukin 1β (IL-1β) levels were investigated using an ELISA technique. The synthesis of interleukins in the liver was elevated in the CD group. Interestingly, SP treatment significantly reduced interleukin production in the liver induced by CD treatment (Figure 6A,B).

## 4. Discussion

The main purpose of this study is to demonstrate, for the first time, the potential therapeutic action of SP against obesity, liver oxidative damage, inflammation, and hepatic steatosis induced by a cafeteria diet in rats.

We first showed the chromatographic analysis of spirulina using HPLC-DAD-ESI-MS/MS, revealing the existence of six phenolic compounds, particularly flavonoids and phenolic acids. Indeed, phenolic acids constitute the major class of these compounds, which are non-flavonoid compounds [20,32]. Additionally, flavonoids include a very wide range of natural phenolic compounds, with nearly 6500 flavonoids categorized into 12 classes [44,45]. These phenolic compounds have been widely studied for their protectiveeffect against various diseases [46,47]. Specifically, flavonoids and phenolic acids are attributed to various effects, including antitumor [48], anti-inflammatory [49], anti-radical, antibacterial, analgesic, anti-allergic, and hepatoprotective effects [50].

Physiologically, we demonstrated that SP treatment reduces the obesity induced by a cafeteria diet, as evidenced by a significant change in body weight, liver enlargement, and a strong accumulation of abdominal fat compared to the standard diet group, which as in agreement with several previous studies [17,30,51]. The obesity induced by the cafeteria diet results from hyperphagia induced by nutritional factors, and is marked by the deposition of abdominal fat into epididymal, perirenal, mesenteric, and retroperitoneal white adipose tissues [17]. CD is a high-fat and high-calorie diet associated with adipose tissue accumulation and weight gain in both humans and rats [52,53]. SP treatment protects against this increase in body weight and hepatic weight and the accumulation of adipose tissue, which may be related to the major quantity of C-phycocyanin in SP [54]. In a similar context, obesity induced by a CD can be attenuated by several medicinal plants, such as lotus (*Nelumbo nucifera*) [55], ginseng [56], and bael (*Aegle marmelos*) [57], as well as by several microalgae [58], including *Phaeodacty lumtricornutum* [59] and *Dunaliella salina* (*D. salina*) [60].

Obesity is not merely a weight problem, it also leads to numerous diseases such as hepatic steatosis [61], renal inflammation [62], atherosclerosis, and endothelial dysfunction [63]. In this context, our study revealed that consuming a CD for two months induced dyslipidemia, as evidenced by the increased triglycerides and cholesterol in plasma and liver tissue, the elevated LDL cholesterol, and the reduction in HDL cholesterol in plasma. Indeed, obesity has been linked to hepatotoxicity due to elevated levels of lipid compounds. The liver overproduces triglycerides and cholesterol, which are transferred to ApoB-100 in the endoplasmic reticulum with the MTP protein (microsomal transfer protein) to form VLDL (very low-density lipoprotein) [64,65], while HDL in the liver is used to transport cholesterol and free fatty acids for their degradation [64]. SP supplementation significantly reduced the risk of dyslipidemia in the plasma and the liver, consistent with previous human studies. Indeed, the intake of 4.5 g of SP for three months reduced TG, TC, and LDL levels in patients with fatty liver disease [66,67].

Additionally, our study demonstrated that a CD caused a significant increase in glycemia, ASAT, ALAT, and direct bilirubin levels. Similar findings have been reported in previous studies [68]. Elevated levels of ASAT and ALAT are indicators of liver steatosis development [68], while high bilirubin amounts predispose the liver to cirrhosis and other hepatic function problems [69]. Importantly, SP treatment effectively regulates glycemia [70] and reduces ASAT, ALAT, and bilirubin levels, suggesting that microalgae might have a preventive effect on liver dysfunction [71,72].

On the other hand, our study confirmed that obesity and oxidative stress are closely linked; this was demonstrated by an increased lipid peroxidation, decreased antioxidant enzyme activities, such as SOD, CAT, and GPx, as well as depletions in non-enzymatic antioxidants such as the -SH group and GSH. These data align with previous reports indicating that liver steatosis is associated with adipose tissue ROS production [17,73,74]. Indeed, the mechanisms underlying non-alcoholic fatty liver disease (NAFLD) are induced by a perturbation in the mitochondrial metabolism, leading to the overproduction of ROS in the liver [73]. On the other hand, our results indicated that a CD was linked to harmful effects to the hydrogen peroxide (H_2_O_2_), hydroxyl radicals (OH•), and superoxide anions (O_2_.^−^) in liver tissue. However, NADPH oxidase expressed in visceral adipose tissue might be responsible for the high generation of hydrogen peroxide [13,75], leading to the production of superoxide anions and the subsequent generation of hydroxyl radicals, the most toxic ROS in oxidative stress [76,77]. This has a significant impact on the pathophysiology of obesity [13]. This link between obesity and oxidative stress has been well demonstrated in both human studies [78] and rodent models [55]. Subchronic treatment with SP significantly protected against hepatic oxidative stress. Several studies have demonstrated the antioxidant capacity of SP to reduce ROS overproduction in the liver [79,80]. According to the findings of Ready et al., SP protects against the depletion in antioxidant enzyme activity [81,82]. The antioxidant activity of spirulina is linked to various biologically active components, including C-phycocyanin, α-tocopherol, β-carotene, phenolic compounds, and phycobiliprotein [54,79,83]. In addition, SP contains superoxide dismutase, which reduces the overproduction of free radicals [84]. Our findings clearly suggest that the inhibition of hyperlipidemia-induced lipid peroxidation in rat livers was related to the ability of the phenolic compounds present in SP to inhibit hepatic oxidative stress [85]. It has been reported that various flavonoids and phenols have a protective effect on liver damage due to their antioxidant activity [20]. Hepatic oxidative stress induced by obesity has been shown to be inhibited by many plants, such as chamomile (*Matrica riarecutita* L.) [17] and *Flaxseed* [86], as well as microalgaesuch as *Chlorella vulgaris* [87] and *Odontella aurita* [88].

Concerning the histological examination, hepatic macrovacuoles were observed, characterized by lipid globules confining the nucleus at the cell periphery. Hepatic steatosis is associated with the accumulation of triglycerides in the cytoplasm of hepatocytes. It is characterized by hepatic injury and necrosis [89,90], which were clearly identified in our histological study. Treatment with SP effectively protected against the initiation of hepatic steatosis, which corroborates previous results on the same hepatoprotective mechanism [91]. In fact, SP contains a wide range of bioactive compounds, including C-phycocyanin and β-carotene, which are both powerful antioxidants and anti-inflammatory agents [92,93]. Additionally, we showed that a CD was linked to detrimental effects on pro-inflammatory cytokines such as TNFα and IL-1β. Obesity is also associated with the chronic inflammation of adipose tissue. Indeed, M1 phenotype macrophages produce numerous pro-inflammatory cytokines, including TNFα, IL-1β, and IL-6, which are associated with T lymphocyte activation in adipose tissue [94]. More importantly, spirulina contains bioactive compounds that exert beneficial effects by decreasing the TNFα and IL-1β levels and the secretion of cytokines such as TNFα [95]; in addition, phycocyanin inhibits enzymes involved in the production of inflammatory molecules such as lipoxgenase (LOX) [96] and suppresses the expression of iNOS and COX-2 [79,97,98]. SP also contains β-carotene, which has potent antioxidant and anti-inflammatory properties [92,99] and inhibits the transcription of pro-inflammatory cytokines such as IL-1β, IL-6, and IL-12 in macrophage cell lines [99].

## 5. Conclusions

In this study, we demonstrated for the first time that SP has a very powerful effect against cafeteria-diet-induced obesity and hepatic steatosis. This effect can be contributed, firstly, to its anti-inflammatory properties and also to its potent ROS scavenging activity. These properties position SP as a promising remedy against obesity and hepatic steatosis. Spirulina has significant potential as a potent anti-inflammatory agent which can be exploited in the future for the mitigation of other obesity-associated diseases and injuries.

## Figures and Tables

**Figure 1 medicina-59-01823-f001:**
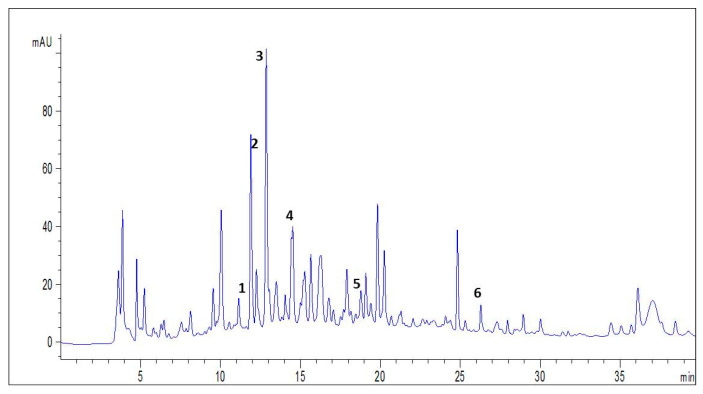
Chromatograms registered with a UV–vis detector at 360 nm for spirulina methanolic extract.

**Figure 2 medicina-59-01823-f002:**
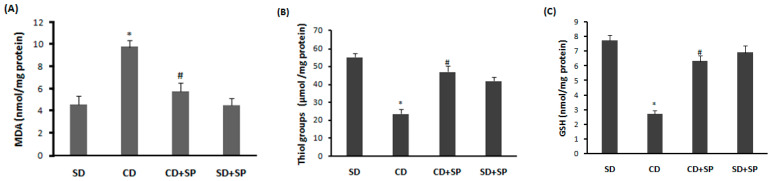
Effect of spirulina (SP) and a cafeteria diet (CD) on hepatic lipoperoxidation (**A**), sulfhydryl groups (**B**), and reduced glutathione (**C**) levels. Rats were fed a standard diet (SD) or a cafeteria diet (CD) and treated with spirulina (SP 500 mg/kg, *b.w.*, *p.o*.) for 8 weeks. Data are expressed as mean S.E.M. (n = 8). ***: *p* < 0.05 compared to control group and *^#^*: *p* < 0.05 compared to CD group.

**Figure 3 medicina-59-01823-f003:**
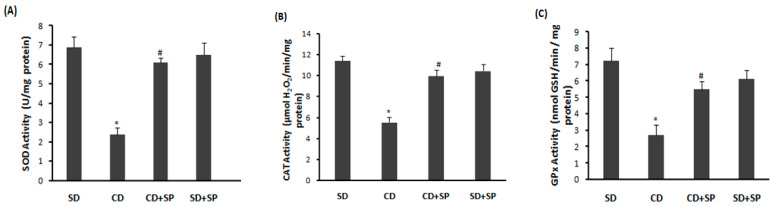
Effect of spirulina (SP) and a cafeteria diet (CD) on hepatic antioxidant enzyme activities: CAT (**A**), SOD (**B**), and GPx (**C**). Rats were fed a standard diet (SD) or a cafeteria diet (CD) and treated with spirulina (SP 500 mg/kg, *b.w*, *p.o.*) for 8 weeks. Data are expressed as mean S.E.M. (n = 8). *: *p* < 0.05 compared to control group and *^#^*: *p* < 0.05 compared to CD group.

**Figure 4 medicina-59-01823-f004:**
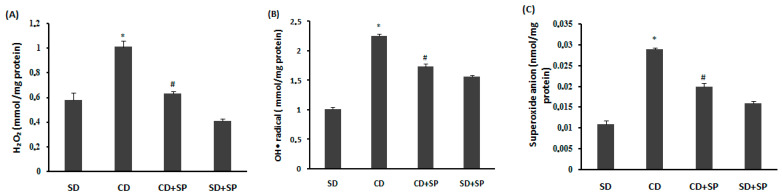
Effect of spirulina (SP) and a cafeteria diet (CD) on hepatic hydrogen peroxide(**A**), hydroxyl radical (**B**), and superoxide anion (**C**) levels.Rats were fed a standard diet (SD) or a cafeteria diet (CD) and treated with spirulina (SP 500 mg/kg, *b.w.*, *p.o*.) for two months. Data are expressed as mean S.E.M. (n = 8). ***: *p* < 0.05 compared to control group and *^#^*: *p* < 0.05 compared to CD group.

**Figure 5 medicina-59-01823-f005:**
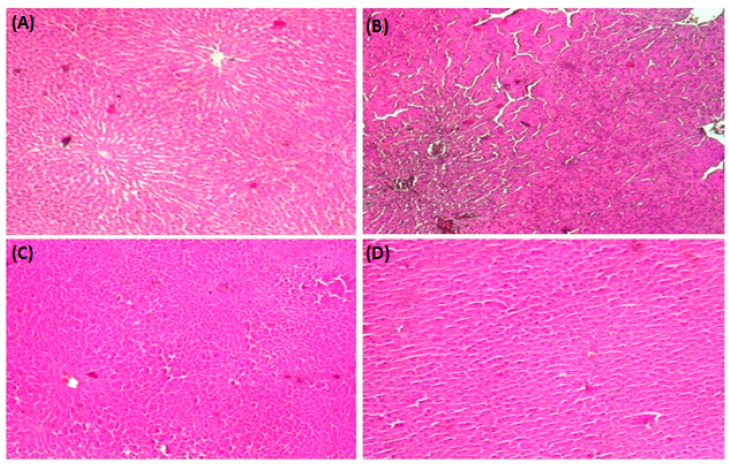
Hepatic histology showing the protective effect of spirulina (SP) on cafeteria diet (CD)-induced histological alterations in livers. Rats were fed a standard diet (SD) or a cafeteria diet (CD) and treated with spirulina (SP 500 mg/kg, *b.w.*, *p.o*.) for 8 weeks. (**A**,**E**) control rats were fed SD; (**B**,**F**) rats were fed CD; (**C**,**G**) CD+ SP (500 mg/k, *b.w.*, *p.o*.), and (**D**,**H**) SD + SP (500 mg/kg, *b.w.*, *p.o*.) (magnification ×10; ×40).

**Figure 6 medicina-59-01823-f006:**
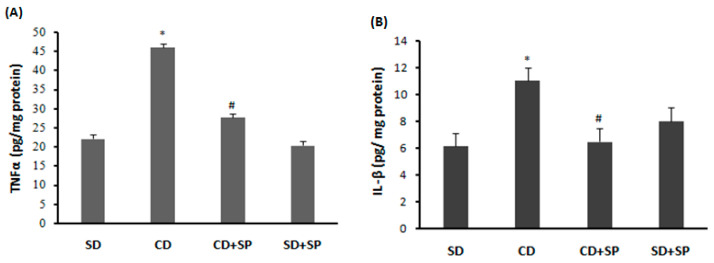
Effect of spirulina (SP) and a cafeteria diet (CD) on hepatic TNFα (**A**) and IL1-β (**B**) levels. Rats were fed a standard diet (SD) or a cafeteria diet (CD) and treated with spirulina (SP 500 mg/kg, *b.w.*, *p.o*.) for two months. Data are expressed as mean S.E.M. (*n* = 8). *: *p* < 0.05 compared to control group and ^#^: *p* < 0.05 compared to CD group.

**Table 1 medicina-59-01823-t001:** Retention time (Rt), molecular mass spectral data, and identification of phenolic compounds in spirulina methanolic extract.

NO	Compounds	Molecular Formula	Molecular Mass	[M − H] −*m/z*	Retention Time (min)	% Composition
1	Resorcinol	C_6_H_6_O_2_	110	109	11,554	26.14
2	Chlorogenic Acid	C_16_H_18_O_9_	354	353	11,996	13.34
3	Catechin	C_15_H_14_O_6_	290	289	12,561	26.48
4	Syringic Acid	C_9_H_10_O_5_	198	197	14,609	5.59
5	Sinapic Acid	C_11_H_12_O_5_	170	169	18,840	18.33
6	Quercetin	C_15_H_10_O_7_	302	301	26,161	10.09

**Table 2 medicina-59-01823-t002:** Effect of spirulina (SP) and a cafeteria diet (CD) on body, liver, and total abdominal fat, weights, and food intake. Rats were fed a standard diet (SD) or a cafeteria diet (CD) and treated with SP (500 mg/kg, *b.w.*, *p.o*.) for 8 weeks.

	SD	CD	CD + SP	SD + SP
**Final Body weight (g)**	268.16 ± 3.73	301.13 ± 7.31 *	270.87 ± 3.27 ^#^	261.61 ± 2.84
**Body weigh gain (g)**	26.37 ± 2.01	52 ± 2.14 *	23.5 ± 2.14 ^#^	16.71 ± 1.98 *
**Liver weight (g)**	5.01 ± 1.11	9.22 ± 0.29 *	7.73 ± 0.23 ^#^	6.53 ± 0.34 ^#^
**Total abdominal fat** **weight (g)**	4.21 ± 0.24	13.49 ± 1.92 *	8.57 ± 0.44 ^#^	4.78 ± 0.17^#^
**Food intake (g)**	110.29 ± 8.18	140.94 ± 4.79 *	124.11 ± 4.96 ^#^	111.97 ± 4.68 ^#^

Data are expressed as mean S.E.M. (n = 8). ***: *p <* 0.05 compared to control group and ^#^: *p* < 0.05 compared to CD group.

**Table 3 medicina-59-01823-t003:** Effect of spirulina(SP) and a cafeteria diet (CD) on plasma and liver lipid profiles. Rats were fed a standard diet (SD) or a cafeteria diet (CD) and treated with spirulina (SP 500 mg/kg, *b.w.*, *p.o.*) for two months.

Pretreatment	TC (mM/L)		TG (mM/L)		HDL-C (mM/L)		LDL-C (mM/L)	
	Plasma	Liver	Plasma	Liver	Plasma	Liver	Plasma	Liver
**SD**	0.95 ± 0.06	1.18 ± 0.08	0.62 ± 0.06	2.05 ± 0,21	0.55 ± 0.03	0.61 ± 0.06	0.52 ± 0.05	0.62 ± 0.06
**CD**	1.83 ± 0.04 *	2.16 ± 0.15 *	1.03 ± 0.07 *	3.85 ± 0.36 *	0.33 ± 0.03 *	0.54 ± 0.03	0.81 ± 0.04 *	0.66 ± 0.05
**CD + SP**	1.14 ± 0.03 ^#^	1.34 ± 0.14 ^#^	0.65 ± 0.05 ^#^	2.46 ± 0.24 ^#^	0.51 ± 0.03 ^#^	0.59 ± 0.03	0.57 ± 0.02 ^#^	0.63 ± 0.04
**SD + SP**	1.01 ± 0.11 ^#^	1.19 ± 0.06 ^#^	0.62 ± 0.07 ^#^	2.09 ± 0.22 ^#^	0.57 ± 0.02 ^#^	0.60 ± 0.01	0.49 ± 0.03 ^#^	0.54 ± 0.04

Data are expressed as mean S.E.M. (n = 8). *: *p* < 0.05 compared to control group and ^#^: *p* < 0.05 compared to CD group.

**Table 4 medicina-59-01823-t004:** Effect of spirulina (SP) and a cafeteria diet (CD) on biochemical parameter changes in obese rats. Rats were fed a standard diet (SD) or a cafeteria diet (CD) and treated with spirulina (SP 500 mg/kg, *b.w.*, *p.o*.) for two months.

	GLY (Mm/L)	ASAT (U/L)	ALAT (U/L)	Bilirubin D (µM/L)
**SD**	5.53 ± 0.20	147.66 ± 6.25	51.5 ± 4.35	0.16 ± 0.04
**CD**	7.89 ± 0.12 *	229.71 ± 12.43 *	92.85 ± 2.94 *	0.27 ± 0.02 *
**CD + SP**	6.42 ± 0.21 ^#^	166.83 ± 10.38 ^#^	57.33 ± 2.13 ^#^	0.17 ± 0.02 ^#^
**SD + SP**	6.04 ± 0.19 ^#^	156.66 ± 15.32 ^#^	50.5 ± 2.66 ^#^	0.14 ± 0.05 ^#^

Data are expressed as mean S.E.M. (n = 8). ***: *p* < 0.05 compared to control group and ^#^: *p* < 0.05 compared to CD group.

## Data Availability

The data presented in this study are available on request from the corresponding author.

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
