# Peer review of "Chromatographic Analyses of Spirulina (Arthrospira platensis) and Mechanism of Its Protective Effects against Experimental Obesity and Hepatic Steatosis in Rats"

_medicina, 2023, doi:10.3390/medicina59101823_

Round 1

Reviewer 1 Report

This paper aimed to investigate chromatographic analyzes of spirulina (Arthrospira platensis) and mechanism of its protective effects against experimental obesity and hepatic steatosis in rats, which was of general interest to medicina. The topic is interesting and finding is novel. However, the paper should be considerably improved. I have listed some comments below:

1.       Please provide the ethical license number for animal experiments.

2.       Please provide the feeding humidity of experimental animals.

3.       The authors stated that during the feeding of experimental animals, there were 5 animals in each cage and 8 animals in each group. How did the author achieve it? Please explain.

4.       Please provide specific methods for the execution of experimental animals to meet the needs of animal ethics.

5.       Line 113. 10000g? Please check whether there are parameter errors here, and provide references to support it.

6.       Line 123, 163. Please provide the manufacturer and model of the commercially kits.

7.       Figure 4C. There is redundant "de" in the vertical coordinate. Please correct the problem here.

8.       Line 256. “H and E” should be “H&E”.

9.       Line 272. TNF-α not classified as an interleukin. Suggest changing the title from liver interleukins levels to liver cytokine levels.

10.   It is recommended to update the references to assess the latest research progress in this field.

11.   The manuscript would benefit from review by an experienced (scientific) English writer.

Moderate editing of English language required.

Author Response

Thank you for your positive response about our manuscript (medicina-2618949). I would like to thank again all the reviewers who contributed to revise our manuscript to become worth for publication. Please note that all the recommendations suggested by the reviewers have been followed point by point in the revised manuscript. All modifications added in the revised manuscript (medicina-2618949) were highlighted in blue colour. Enclose with this cover letter an explanatory report and a revised manuscript. 

Reviewer: 1

This paper aimed to investigate chromatographic analyzes of spirulina (Arthrospira platensis) and mechanism of its protective effects against experimental obesity and hepatic steatosis in rats, which was of general interest to medicina. The topic is interesting and finding is novel. However, the paper should be considerably improved. I have listed some comments below:

  1. Please provide the ethical license number for animal experiments.

OK, the ethical license number for animal experiments has been added in "Animals and treatment" section.

  1. Please provide the feeding humidity of experimental animals.

OK, the feeding humidity of experimental animals has been added in "Animals and treatment" section.

  1. The authors stated that during the feeding of experimental animals, there were 5 animals in each cage and 8 animals in each group. How did the author achieve it? Please explain.

Sorry for this inadvertence, the rats were separated into four groups, each with eight rodents, at

the rate of two cages for each treatment group (4 rats per cage). See modification in the Animals and treatment" section.

  1. Please provide specific methods for the execution of experimental animals to meet the needs of animal ethics.

Ok. See modification in the "Animals and treatment" section.

  1. Line 113. 10000g? Please check whether there are parameter errors here, and provide references to support it.

No errors. A reference has been added.

  1. Line 123, 163. Please provide the manufacturer and model of the commercially kits.

Ok, the manufacturer and model of the commercially kits have been added.

  1. Figure 4C. There is redundant "de" in the vertical coordinate. Please correct the problem here.

Sorry for this inadvertence, this figure has been corrected.

  1. Line 256. “H and E” should be “H&E”.

Ok, this line has been corrected according to reviewer suggestion.

  1. Line 272. TNF-α not classified as an interleukin. Suggest changing the title from “liver interleukins levels” to “liver cytokine levels”.

Sorry for this inadvertence, this error has been corrected throughout the manuscript.

  1. It is recommended to update the references to assess the latest research progress in this field.

Ok, the references list has been updated according to reviewer suggestion.

  1. The manuscript would benefit from review by an experienced (scientific) English writer.

All manuscript section has been checked for English language with the help of a scientific English user

Reviewer 2 Report

Dear Author, I reviewed the manuscript (medicina-2618949) entitled Chromatographic analyzes of spirulina (Arthrospira platensis) and mechanism of its protective effects against experimental obesity and hepatic steatosis in rats. This manuscript presents relevant information about spirulina's bioactive properties. However, some sections of the presented data can be improved. For this reason, I consider that this manuscript needs minor changes to be considered for publication in this journal. 

Additional comments.

Highlight the advantages of using spirulina by its bioactivity.

Check paragraphs extension in this manuscript.

Include bibliographical references in some methodologies of the manuscript. 

Include an experimental design that contains statistical factors and variables of response in the statistical analyses applied to the findings of this research.

Include a possible anti-inflammatory mode of action of spirulina bioactive compounds.

Compare the obtained findings with similar assays where other microalgae were used to study obesity and hepatic steatosis. 

Include future trends to keep working with the obtained data. 

Try to conclude with a general statement of the most relevant part of this study.

Author Response

Thank you for your positive response about our manuscript (medicina-2618949). I would like to thank again all the reviewers who contributed to revise our manuscript to become worth for publication. Please note that all the recommendations suggested by the reviewers have been followed point by point in the revised manuscript. All modifications added in the revised manuscript (medicina-2618949) were highlighted in blue colour. Enclose with this cover letter an explanatory report and a revised manuscript. 

Reviewer(s)' Comments to Author: 

Reviewer: 2

I reviewed the manuscript (medicina-2618949) entitled Chromatographic analyzes of spirulina (Arthrospira platensis) and mechanism of its protective effects against experimental obesity and hepatic steatosis in rats. This manuscript presents relevant information about spirulina's bioactive properties. However, some sections of the presented data can be improved. For this reason, I consider that this manuscript needs minor changes to be considered for publication in this journal.

Additional comments.

  1. Highlight the advantages of using spirulina by its bioactivity.

Ok, particular attention was paid to the advantages of using spirulina by its bioactivity. See modification in the text.

  1. Check paragraphs extension in this manuscript.

Ok, checks have been carried out.

  1. Include bibliographical references in some methodologies of the manuscript.

Ok, the references list has been updated according to reviewer suggestion.

  1. Include an experimental design that contains statistical factors and variables of response in the statistical analyses applied to the findings of this research.

Statistical factors and variables have been improved in our study

  1. Include a possible anti-inflammatory mode of action of spirulina bioactive compounds.

Ok, models and examples have been added. See modification in the text.

  1. Compare the obtained findings with similar assays where other microalgae were used to study obesity and hepatic steatosis.

Ok, comparisons were made. See modification in the text.

  1. Include future trends to keep working with the obtained data.

Ok, our future projects have been added. See modification added in "Conclusion" section.

  1. Try to conclude with a general statement of the most relevant part of this study.

Ok, see modification added in "Conclusion" section.

Reviewer 3 Report

Dear authors,

I consider your manuscript entitled "Chromatographic analyzes of spirulina (Arthrospira platensis) and mechanism of its protective effects against experimental obesity and hepatic steatosis in rats" very interesting. However, there are just some points that need to be clarified.

  • Firstly, you should provide a brief definition of hepatic steatosis and, then, at least mention the mechanisms that link oxidative stress with obesity, inflammation, and hepatic damage (NAFLD). See doi.org/10.1155/2018/9547613, doi.org/10.3390/livers2010003, and doi.org/10.1016/j.freeradbiomed.2020.02.025.
  • Lines 53–56 add recent references about supplementation with spirulina (doi:10.1080/09637486.2022.2137785, doi: 10.3389/fnut.2022)
  • Lines 92–94: Regarding the CD diet, is there a rationale behind its composition? Why did you choose 50% and 50%?

Overall, the article is well written and organised.

Author Response

Thank you for your positive response about our manuscript (medicina-2618949). I would like to thank again all the reviewers who contributed to revise our manuscript to become worth for publication. Please note that all the recommendations suggested by the reviewers have been followed point by point in the revised manuscript. All modifications added in the revised manuscript (medicina-2618949) were highlighted in blue colour. Enclose with this cover letter an explanatory report and a revised manuscript. 

Reviewer(s)' Comments to Author: 

Reviewer: 3

I consider your manuscript entitled "Chromatographic analyzes of spirulina (Arthrospira platensis) and mechanism of its protective effects against experimental obesity and hepatic steatosis in rats" very interesting. However, there are just some points that need to be clarified.

  1. Firstly, you should provide a brief definition of hepatic steatosis and, then, at least mention the mechanisms that link oxidative stress with obesity, inflammation, and hepatic damage (NAFLD). See doi.org/10.1155/2018/9547613, doi.org/10.3390/livers2010003, and doi.org/10.1016/j.freeradbiomed.2020.02.025.

Ok, improvements in the text have been made according to reviewer suggestion.

  1. Lines 53–56 add recent references about supplementation with spirulina (doi:10.1080/09637486.2022.2137785, doi: 10.3389/fnut.2022)

Ok, recent references have been added according to reviewer suggestion.

  1. Lines 92–94: Regarding the CD diet, is there a rationale behind its composition? Why did you choose 50% and 50%?

Ok, explanatory references have been added.

  1. Overall, the article is well written and organised.

Many thanks, all manuscript section has been checked for English language with the help of a scientific English user.

Round 2

Reviewer 1 Report

Authors addressed the previous questions and, by doing so, improved the manuscript.

Reviewer 3 Report

well done